# Mineral Classification of Soils Using Hyperspectral Longwave Infrared (LWIR) Ground-Based Data

**Gila Notesco \*, Shahar Weksler**  **and Eyal Ben-Dor**

Porter School of the Environment and Earth Sciences, Faculty of Exact Sciences, Tel Aviv University,
Tel Aviv 6997801, Israel; tonish@gmail.com (S.W.); bendor@tauex.tau.ac.il (E.B.-D.)
\* Correspondence: gilano@tauex.tau.ac.il

**Abstract:** Soil mineralogy is an important factor affecting chemical and physical processes in the soil. Most common minerals in soils—quartz, clay minerals and carbonates—present fundamental spectral features in the longwave infrared (LWIR) region. The current study presents a procedure for determining the soil mineralogy from the surface emissivity spectrum. Ground-based hyperspectral LWIR images of 90 Israeli soil samples were acquired with the Telops Hyper-Cam sensor, and the emissivity spectrum of each sample was calculated. Mineral-related emissivity features were identified and used to create indicants and indices to determine the content of quartz, clay minerals, and carbonates in the soil in a semi-quantitative manner—from more to less abundant minerals. The resultant mineral content was in good agreement with the mineralogy derived from chemical analyses.

**Keywords:** hyperspectral remote sensing; longwave infrared image; emissivity spectrum; soil mineralogy

## 1. Introduction

Soil is a complex material that is extremely variable in its physical and chemical composition. It consists of weathered rocks of the Earth's crust and is continuously evolving and changing. The upper layer is the most important part of the soil body, containing plant litter, water, roots, micro and macro fauna, and minerals that are formed in situ or transferred to the area by runoff, dust storms, or human activity. The soil surface is exposed to all remote-sensing methods. Hyperspectral remote sensing, especially in the visible–short wavelength infrared region, 0.4–2.5 μm, has been shown to be an invaluable tool for determining and mapping soil properties, using laboratory, airborne, and spaceborne sensors (e.g., [1–6]). However, recent studies have shown that the thermal infrared region, in both the mid wavelength infrared (3–5 μm) and the long wavelength infrared (LWIR, 8–12 μm) regions can provide quantitative information on soil properties, such as texture, carbon and nitrogen content, and pH, especially when large datasets are processed with statistical models (e.g., [7–11]). Soil mineralogy is an important factor in determining its properties, quality, and growth potential, and is extensively used as a diagnostic criterion in comprehensive soil classification. The most common minerals in soils, quartz, clay minerals, and carbonates, present with fundamental spectral features in the thermal infrared, and mainly in the LWIR region due to the fundamental vibration modes of the silicon–oxygen bond (Si–O) in quartz and clay minerals, and the carbon–oxygen bond (C–O) in carbonates. Recently, we presented a procedure that can be applied to hyperspectral LWIR images to calculate surface emissivity, an important variable for mineral mapping, and to identify the dominant minerals, quartz, feldspars, clay minerals, gypsum, and carbonates in rocks [12,13]. Nevertheless, hyperspectral remote sensing in the LWIR region, although used with significant success for the

mapping of the content of minerals on rock surfaces (e.g., [14,15]), has not been fully implemented for the complex and dynamic soil material, and its potential has not yet been fully exploited.

The current study makes use of ground-based hyperspectral LWIR images, acquired with the Telops Hyper-Cam hyperspectral sensor [16], to calculate the emissivity spectra of soil samples, representing the soil's chemical and physical properties, and then to analyze them to identify quartz, clay minerals, carbonates, and their abundance, in each sample. The resultant mineral content was correlated to the chemical elements' abundance and the mineral composition as obtained from chemical analyses.

## 2. Materials and Methods

### 2.1. Soil Samples and Chemical Analyses

Ninety soil samples from the legacy soil spectral library of Israel, representing different formation conditions—climate, origin, and topography—were collected from the surface (0–5 cm depth) at different sites in Israel. The soils were air-dried and gently crushed to a grain size of ≤2 mm. Elemental analysis was carried out using the X-ray fluorescence (XRF) method [17]. X-ray powder diffraction (XRD) patterns of the samples were obtained using a Philips Model 1010 X-ray diffractometer with Fe-filtered CoKa radiation [18], to determine the mineralogy of each soil sample. The abundance of silicon (Si, dominant in quartz and clay minerals), aluminum (Al, dominant in clay minerals) and calcium (Ca, dominant in carbonates), and the mineral content in selected soil samples are given in Table 1.

**Table 1.** Element abundance and mineralogy of soil samples.

| Soil (Symbol, USDA Name) | Element Abundance (%) | | | Mineral Abundance (%) | | |
|---|---|---|---|---|---|---|
| | Si | Al | Ca | Quartz | Clay Minerals [a] | Carbonates [b] |
| E2, Rhodoxeralf | 91.6 | 6.78 | 0.57 | 90 | 5 | 0 |
| E7, Rhodoxeralf | 87.2 | 10.5 | 0.30 | 75 | 15 | 1 |
| C4, Haploxeroll | 67.2 | 12.8 | 7.80 | 55 | 30 | 12 |
| B8, Haploxeroll | 42.0 | 17.8 | 17.3 | 30 | 40 | 31 |
| A3, Rhodoxeralf | 51.5 | 24.4 | 3.37 | 35 | 58 | 2 |
| H2, Xerert | 41.1 | 24.6 | 11.2 | 4 | 67 | 21 |
| K2, Calciorthid | 21.1 | 7.60 | 35.3 | 15 | 0 | 60 |
| O3, Torriorthent | 26.4 | 8.04 | 31.3 | 25 | 10 | 59 |
| P3, Torriorthent | 25.3 | 5.76 | 36.5 | 25 | 7 | 68 |
| H11, Haplargid | 29.7 | 4.44 | 30.1 | 36 | 10 | 54 |
| H14, Xerert | 38.4 | 18.7 | 16.3 | 25 | 45 | 28 |
| S19, Torriorthent | 35.4 | 11.9 | 26.0 | 47 | 10 | 43 |

[a] Clay minerals refer to smectite, illite, and kaolinite. [b] Carbonates refer to calcite (mainly) and dolomite.

### 2.2. Spectral Measurements and Data Analysis

Ground-based images of the 90 soil samples were acquired with the Telops Hyper-Cam, covering the LWIR spectral region (8.0–11.7 μm) with 122 bands and a spectral resolution of 4 cm$^{-1}$. The samples were placed at a distance of about 2 m from the sensor, exposed to the sun (with an outdoor air temperature of ~30 °C) for about an hour, and then the LWIR images (see example in Figure 1a) were taken. The at-sensor radiance at wavelength λ ($L_{s\lambda}$) measured from each soil pixel in the image consisted of two components (as explained in [19]):

$$L_{s\lambda} = \varepsilon_\lambda L_{b\lambda}(T) + (1 - \varepsilon_\lambda)L_{d\lambda} \tag{1}$$

where $\varepsilon_\lambda L_{b\lambda}(T)$ is the surface emission at wavelength λ with $\varepsilon$ as the surface emissivity, and $L_b(T)$ as the radiance from a blackbody at surface temperature T; $(1-\varepsilon_\lambda)L_{d\lambda}$ is the surface reflection of the radiance incident on the surface from the atmosphere at wavelength λ with $L_d$ as the downwelling

radiance. The emissivity spectrum of each pixel was calculated according to Equation (1), with $L_b(T)$ as the fitted tangent blackbody radiation curve, applying the specialized algorithm described in [19], and downwelling radiance ($L_d$) as the radiance measured from a gold plate [20] (Figure 1b). The emissivity spectrum of each soil sample was analyzed to determine its mineral content.

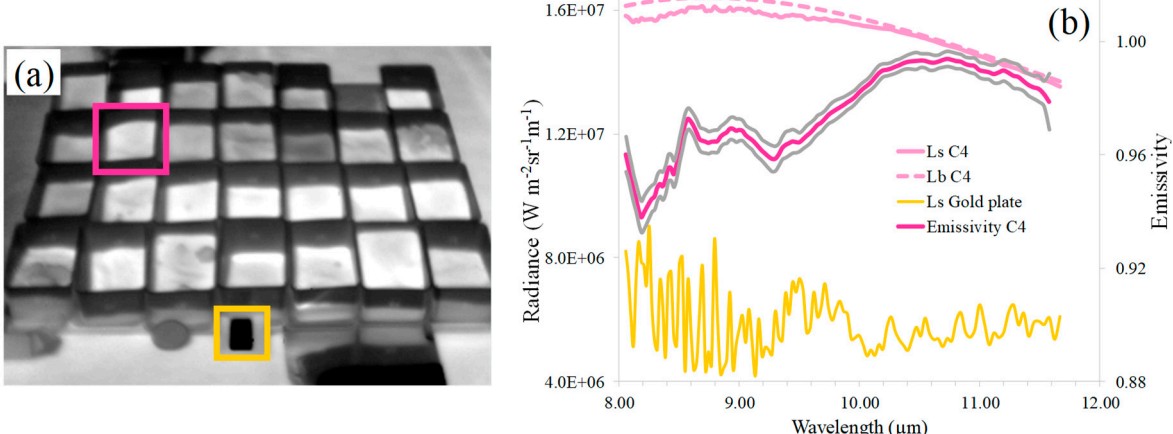

**Figure 1.** (**a**) Longwave infrared (LWIR) image (band 10.62 μm) of soil samples (each placed in a 17 × 22 cm box); soil C4 (pink square), and gold plate (yellow square). (**b**) At-sensor radiance (solid pink curve) and fitted tangent blackbody radiation (dashed pink curve) of soil C4, at-sensor radiance of gold plate (yellow curve) representing the downwelling radiance (left *y*-axis), calculated emissivity of soil C4 (right *y*-axis) with the respective ± standard deviation curves (gray). Each spectrum is the average of tens of pixels in the image.

## 3. Results and Discussion

The content of the minerals quartz, clay minerals and carbonates in each sample was determined based on the absorption features in the emissivity spectrum, as demonstrated in Figures 2 and 3. A triplet-like absorption feature with minima at 8.21 μm, 8.85 μm, and 9.33 μm (Figure 2a) indicates the presence of both quartz and clay minerals (e.g., soil E2). The ratio between the minimum values, and the wavelength of the third minimum, depend on the relative amounts of quartz and clay minerals in the soil. A decrease in the amount of quartz with an increase in the amount of clay minerals reduces the value of the absorption at 8.21 μm and shifts the absorption at 9.33 μm to 9.36–9.56 μm (e.g., soil A3). On the other hand, a minimum in the 8.06–8.12 μm range and/or a noticeable absorption feature between 10.20 μm and 11.40 μm, as demonstrated in Figure 3, indicates the presence of carbonates in the soil. The associated emissivity features of pure minerals are given in Figure 4.

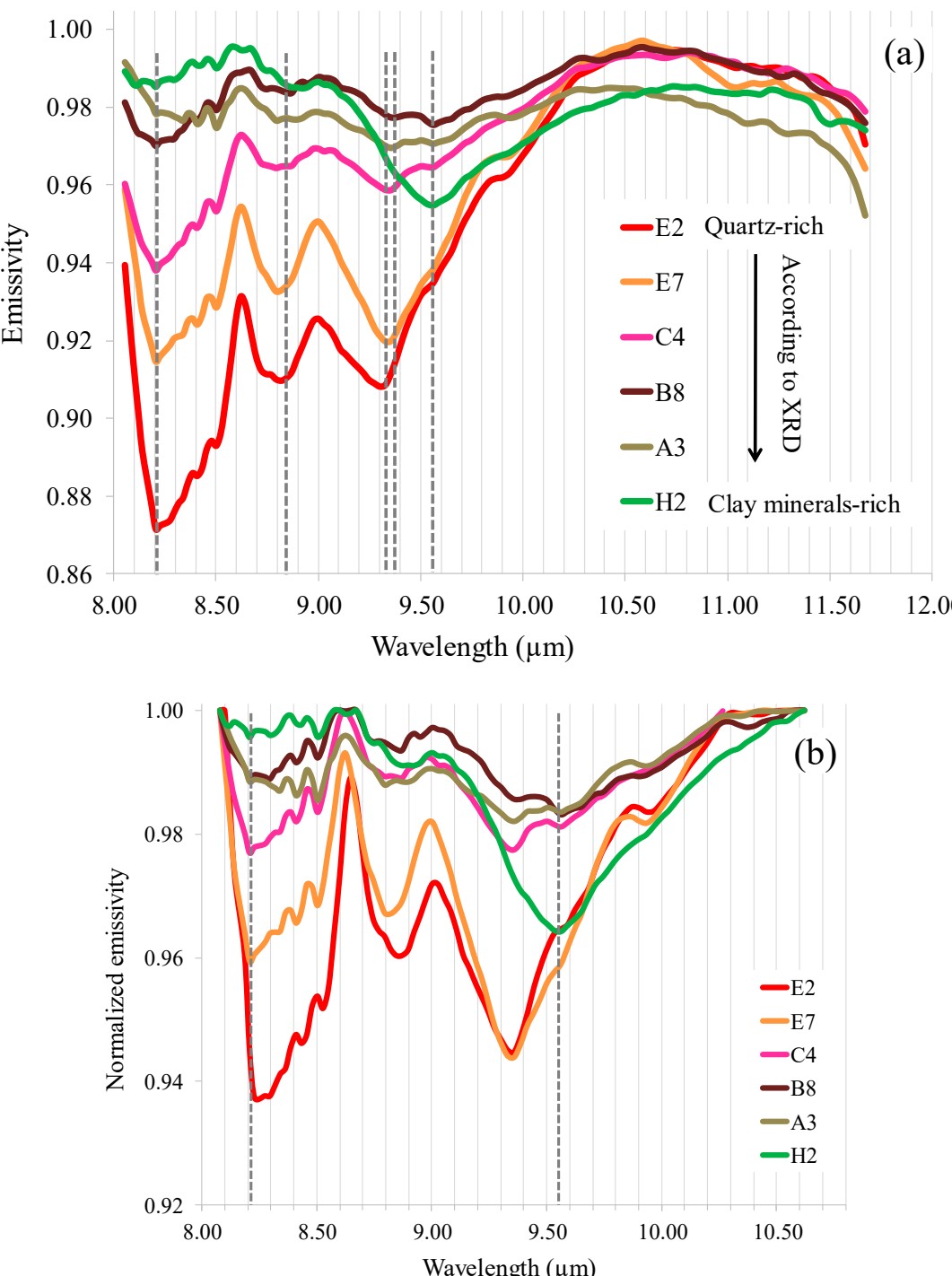

**Figure 2.** (**a**) Emissivity spectra of selected soil samples. Each spectrum is the average of tens of pixels in the calculated emissivity image. (**b**) Normalized emissivity spectra emphasizing indicative absorption features of quartz and clay minerals. Gray dashed lines emphasize the wavelengths mentioned in the text.

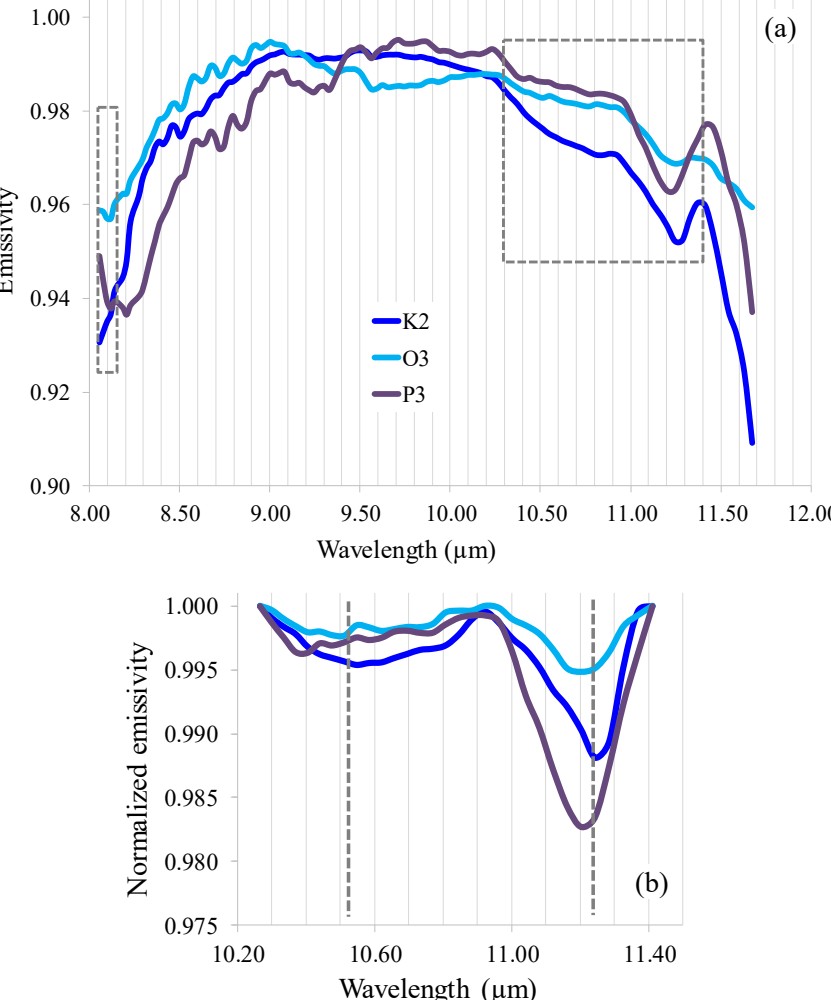

**Figure 3.** (**a**) Emissivity spectra of selected soil samples; gray dashed boxes emphasize indicative absorption features of carbonates. Each spectrum is the average of tens of pixels in the calculated emissivity image. (**b**) Normalized emissivity spectra; gray dashed lines emphasize the wavelengths mentioned in the text.

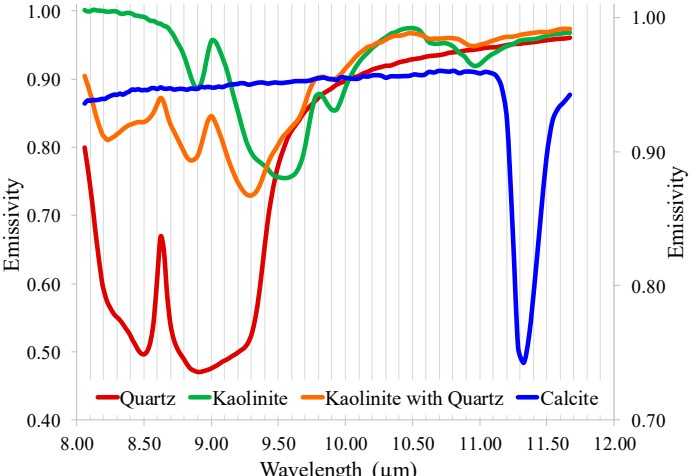

**Figure 4.** Emissivity spectra of pure minerals: quartz, kaolinite, kaolinite with quartz substrate (left *y*-axis) and calcite (right *y*-axis), based on [21] and resampled to the Telops Hyper-Cam spectral configuration.

The aforementioned mineral-related emissivity features were used to create spectral indicants, described in Table 2, to identify the most abundant mineral(s) in each soil sample. These were then used to classify the samples into three soil types: CM (clay minerals as most abundant minerals), C (carbonates as most abundant minerals), and Q (quartz as most abundant mineral).

**Table 2.** Spectral indicants of minerals.

| Most Abundant Mineral | Spectral Indicant |
|:---:|:---:|
| Clay minerals | $N\varepsilon_{\lambda = 9.56\mu m}{}^{a} < N\varepsilon_{\lambda = 8.21\mu m}$ and $N\varepsilon_{\lambda = 8.21\mu m} > 0.98$ |
| Carbonates | $\varepsilon_{\lambda = 8.06\text{-}8.12\mu m}{}^{b} < \varepsilon_{\lambda = 8.21\mu m}$ and/or $N\varepsilon_{\lambda = 11.24\mu m} < 0.995$ with $N\varepsilon_{\lambda = 8.21\mu m} > 0.98$ |
| Quartz | Excluding the above |

[a] $N\varepsilon$ is the normalized emissivity value at the indicated wavelength. [b] $\varepsilon$ is the emissivity value in the indicated wavelength range.

The spectral-based classification fit the XRD analysis results in most (90%) of the soil samples (Figure 5), as well as the elemental analysis results (Figure 6). The Al/Si values of the CM-type soils were larger than most Al/Si values of the Q-type soils (Figure 6a). In general, a larger Al/Si value in the sample indicates a higher concentration of clay minerals than quartz. The abundance of Ca (indicating the concentration of carbonates) in soils classified as a C-type was larger than in soils classified as a CM or Q-type (Figure 6b).

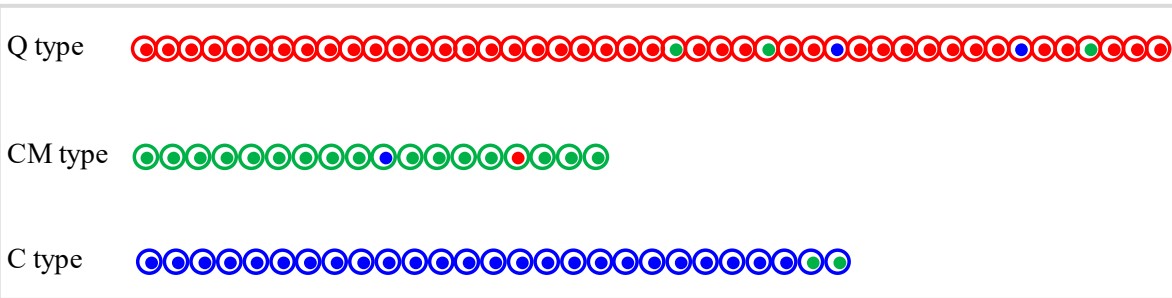

**Figure 5.** Type of each soil sample according to the most abundant mineral(s)—carbonates (blue), clay minerals (green), and quartz (red); each pair of circles (empty and filled) represent a soil sample; the spectral-based classification (empty circle) fits the XRD analysis results (filled circle) where the circle color is identical.

Once the most abundant minerals were determined, we turned to identifying the less abundant minerals in each soil sample using two indices that were created based on their mineral-related emissivity features:

$$\text{SQCMI (Soil Quartz Clay Mineral Index)} = N\varepsilon_{\lambda = 9.56\ \mu m}/(N\varepsilon_{\lambda = 8.21\ \mu m} \times N\varepsilon_{\lambda = 8.85\ \mu m}) \quad (2)$$

and

$$\text{SCI (Soil Carbonate Index)} = N\varepsilon_{\lambda = 11.24\ \mu m} \times N\varepsilon_{\lambda = 10.51\ \mu m}/N\varepsilon_{\lambda = 8.85\ \mu m} \quad (3)$$

where $N\varepsilon$ is the normalized emissivity value at the indicated wavelength.

In general, a larger SQCMI value indicates a higher Si/Al ratio (Figure 7a), suggesting a lower concentration of clay minerals relative to quartz; a smaller SCI value indicates a higher concentration of carbonates in the soil sample (Figure 7b).

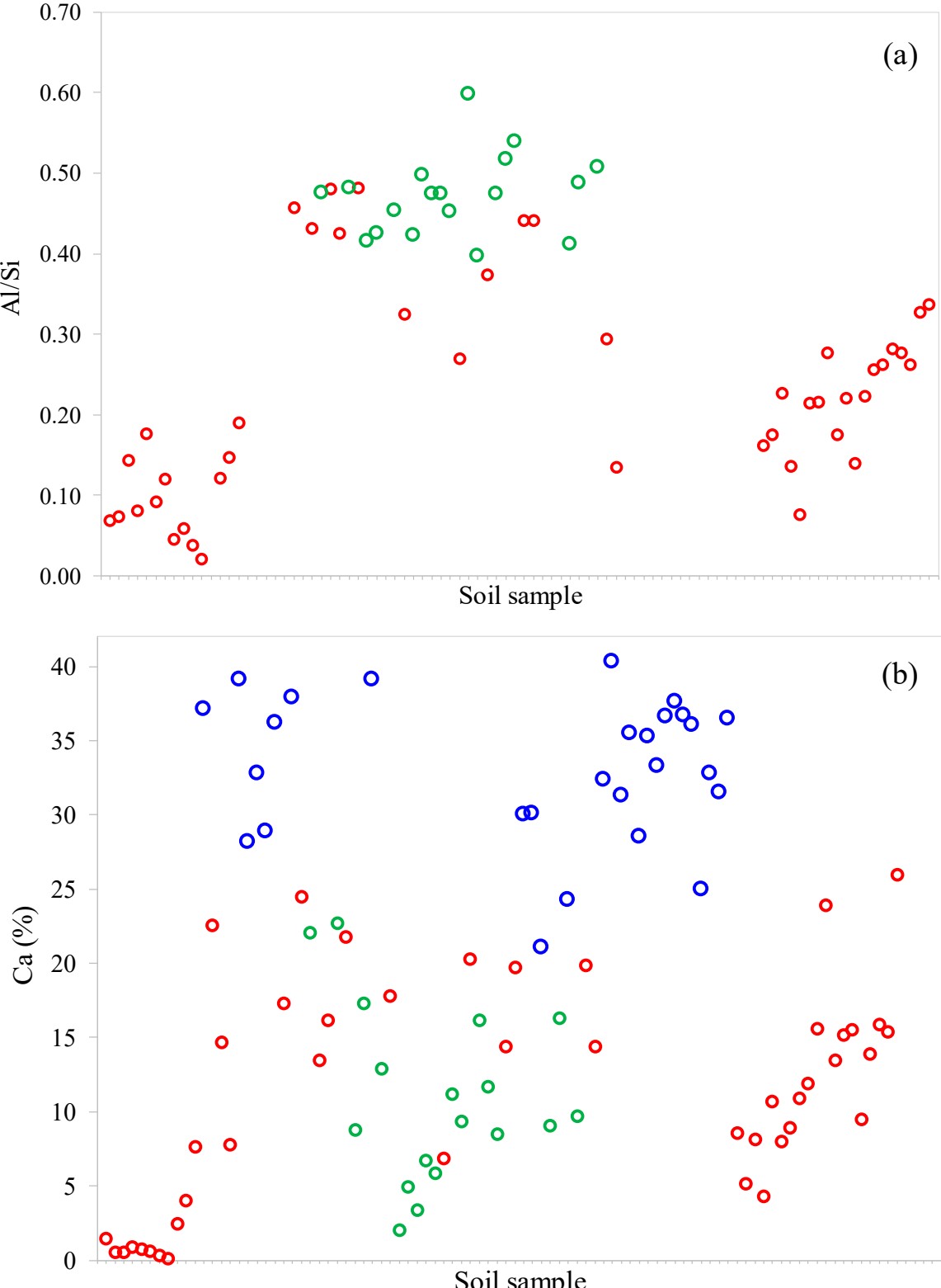

**Figure 6.** The Al/Si value (**a**) and % Ca (**b**) of each soil sample; red circles represent spectral-based Q-type soils, green circles represent spectral-based CM-type soils, and blue circles represent spectral-based C-type soils.

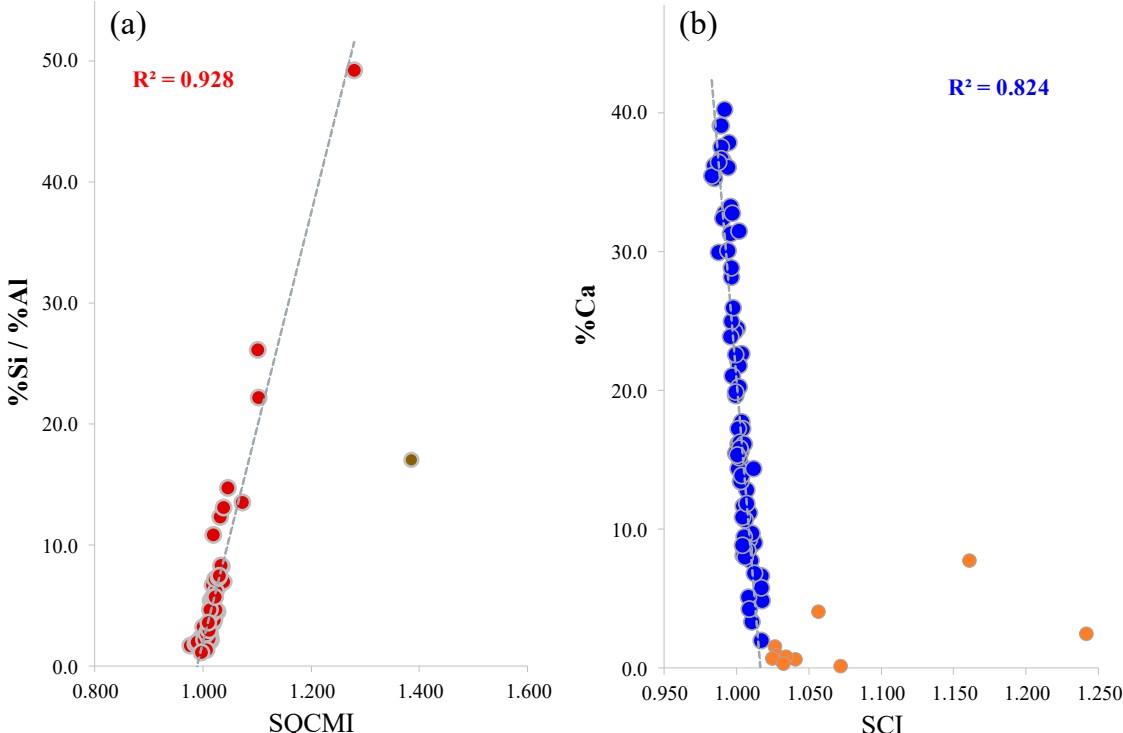

**Figure 7.** (**a**) Correlation between the % Si/% Al ratio and the calculated SQCMI value of all soil samples, excluding one brown–red sandy soil sample (brown circle). (**b**) Correlation between the % Ca values and the calculated SCI values of soil samples, excluding brown–red sandy soils (orange circles), characterized by relatively small amounts of carbonates.

Comparing the index values with the chemical analysis resulted in spectral indicants, described in Table 3, which enabled determining the mineralogy, from more to less abundant, in each soil sample.

**Table 3.** Spectral indicants of the relative amounts of minerals.

| Soil Type | Indicant | Relative Amount of Mineral(s) |
|:---:|:---|:---:|
| Q | SCI < 1.010 | C > CM |
| | $1.010 \leq SCI < 1.020$ and SQCMI > 1.020 | C > CM |
| | SCI > 1.020 | CM > C |
| | SCI > 1.050, SQCMI > 1.200 | no C, no CM |
| CM | Absorption at 8.12 µm and/or SCI < 1.005 | C > Q |
| C | SQCMI > 1.010 with $N\varepsilon_{\lambda = 8.21\mu m} < 0.990$ | Q > CM |

Emissivity spectra of selected soil samples, their indices and mineralogy are presented in Figure 8 and Table 4, respectively.

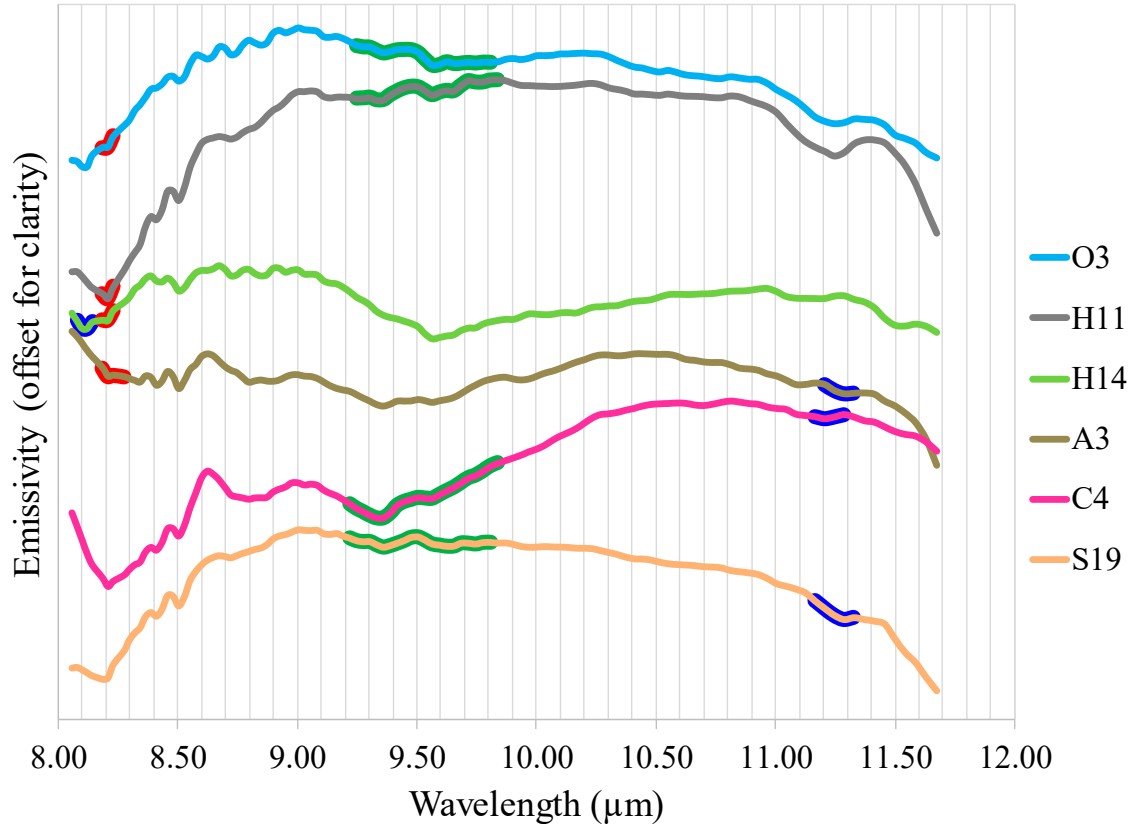

**Figure 8.** Emissivity spectra of selected C-type (O3, H11), CM-type (H14, A3) and Q-type (C4, S19) soil samples; bold curves represent mineral-related features of the less abundant minerals (blue for carbonates, red for quartz and green for clay minerals) in each sample.

**Table 4.** Spectral indicants and resultant mineralogy of selected soil samples.

| Soil | Type | Indicants | Mineralogy (More to Less Abundant) | |
|------|------|-----------|-----------------|---------------|
| | | | **Spectral-Based** | **XRD Analysis** |
| E2 | Q | SQCMI = 1.072, SCI = 1.041 | Q CM C | Q CM |
| E7 | Q | SQCMI = 1.033, SCI = 1.033 | Q CM C | Q CM C |
| C4 | Q | SQCMI = 1.015, SCI = 1.010 | Q CM C | Q CM C |
| B8 | CM | SCI = 1.004 | CM C Q | CM C Q |
| A3 | CM | SCI = 1.010 | CM Q C | CM Q C |
| H2 | CM | SCI = 1.008, absorption at 8.12 µm | CM C Q | CM C Q |
| K2 | C | SQCMI = 1.004 | C CM Q | C Q CM |
| O3 | C | SQCMI = 1.000 | C CM Q | C Q CM |
| P3 | C | SQCMI = 1.020 | C Q CM | C Q CM |
| H11 | C | SQCMI = 1.017, $N\varepsilon_{\lambda\ =\ 8.21\,\mu m} = 0.983$ | C Q CM | C Q CM |
| H14 | CM | SCI = 1.002, absorption at 8.12 µm | CM C Q | CM C Q |
| S19 | Q | SQCMI = 1.012, SCI = 0.997 | Q C CM | Q C CM |

The spectral-based full mineralogy of most (75%) soil samples fit the XRD analysis results (Figure 9).

Overall, the indicants enabled the determining of the mineral content in the soil in a semi-quantitative manner, from more to less abundant minerals.

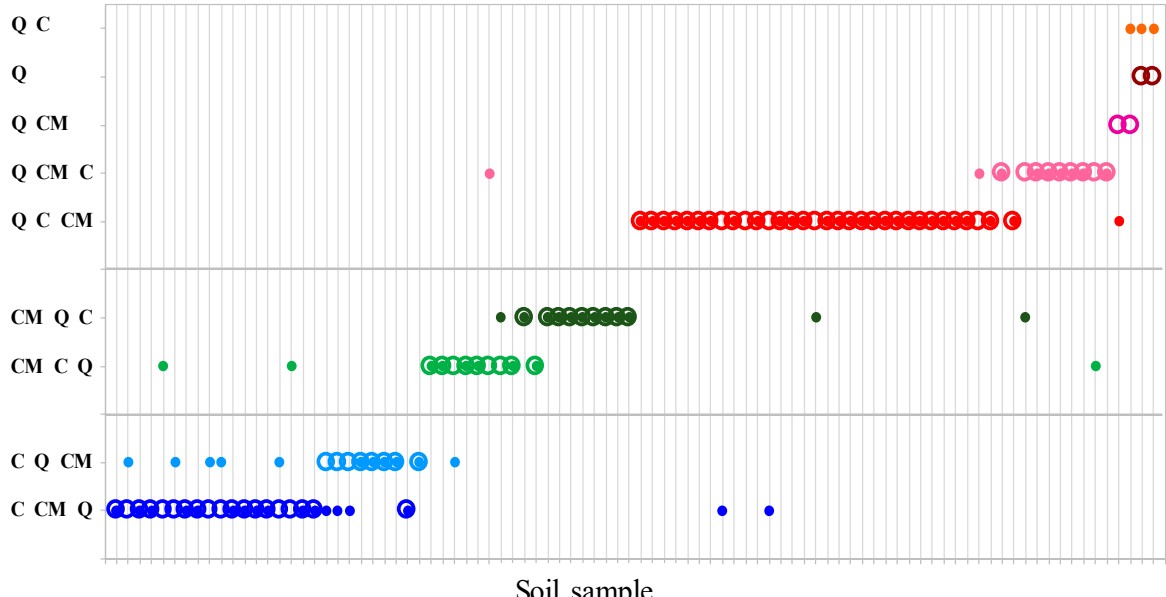

**Figure 9.** Mineralogy (from more to less abundant, *y*-axis) of each soil sample; spectral-based classification (empty circles) and XRD analysis results (filled circles).

## 4. Conclusions

The current study presents a procedure for determining the soil mineralogy using ground-based hyperspectral LWIR images. The emissivity spectra of 90 soil samples were calculated and analyzed to identify the most common minerals in soils, quartz, clay minerals, and carbonates, and their relative abundance in each sample. Identifying the mineral-related emissivity features and their relative intensities enabled the creation of indicants to identify the most abundant mineral(s) in each soil sample and to classify it as a Q-, CM-, or C-type soil. The spectral-based soil type classification fit the results of the XRD and elemental analyses in most (90%) of the soil samples. The relative amounts of the less abundant minerals in each soil sample were determined using two created indices—SQCMI and SCI, resulting in a semi-quantitative mineralogy determination. The spectral-based full mineralogy of most (75%) of the soil samples fit the XRD analysis results. The presence of organic matter, fertilizers, crude oil, and other contaminating materials should be studied in terms of whether and how they affect the indicant and index values, and therefore, the determination of the soil mineralogy.

The presented procedure can be used to study the effects of different processes, e.g., Aeolian processes, aquaturbation, and fire events on the soil surface mineralogy. For a known soil surface, an increase in the SQCMI value may indicate an increase in the amount of quartz, relative to clay minerals, and a decrease in the SCI value may indicate an increase in the amount of carbonates.

As hyperspectral technology in the thermal LWIR region is becoming established in the field of terrestrial mapping, it is important to study its potential for soil mapping. The ground-based procedure should be implemented for airborne hyperspectral LWIR data, and the ability to detect quartz, clay minerals, and carbonates, and to map their abundance in the soil surface and monitor mineralogical changes on a regional scale and should be studied further.

**Author Contributions:** Conceptualization, G.N. and E.B.-D.; methodology, formal analysis, validation, writing—original draft preparation and editing, G.N.; resources, writing—review, supervision, E.B.-D.; sensor operation and data acquisition, S.W.

**Funding:** This research was funded by the Israel Ministry of Science, Technology and Space, grant number 68740 and the Israel Science Foundation, grant number 1395/15.

**Conflicts of Interest:** The authors declare no conflict of interest.

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
