# Peer review of "Mineral Classification of Soils Using Hyperspectral Longwave Infrared (LWIR) Ground-Based Data"

_remotesensing, doi:10.3390/rs11121429_

Round 1

Reviewer 1 Report

This work represents an important contribution to the knowledge and systematization procedures of the spectral behaviour of soils in the LWIR region associated with their mineralogy, opening the door for soil interpretation and mapping in this spectral region from hyperspectral images.

Author Response

The authors would like to thank the reviewer.

Reviewer 2 Report

Dear Authors,

you focus on very important topic, you have a good set of methods, but the manuscript is significantly too general. It looks like a report. You have to add much more details to all parts of the manuscript. The most important is to:

add details to the Introduction, you need to present details of references (please, eliminate the citation trains),

add a research schema focusing on data acquisition, processing and an accuracy assessment.

you need to add more details to the Results.

Your Discussion doesn't exists, you need to find references to compare your scored results with outcomes achieved by other researchers, so please add references to discussion.

Much more details you can find in the attached manuscript.

All the best

Reviewer

Author Response

The authors would like to thank the reviewer. Our response to the reviewer's comments are in red color. 

1. You use citation trains. Please, indicate an input of references, why you seleceted these citation? Please add details. References [1-6] and [7-11] were cited as examples to the text.  

2. The Introduction is significantly too short, please, add much more details, to present a theoretical background of your topic. The type of the manuscript is “Letter”, complying the journal’s instructions: “… typically brief (less than 10 pages) explanations of a single concept, technique, or study…contain less information than an article and are suitable for rapid dissemination of results.” Therefore, we prefer to keep manuscript’s sections “short and sweet”.

3. Please, add a research schema. A concise description of research methodology was given in the last paragraph of the introduction.

4. Please, add much more details of data acquisition and processing. Necessary details were given in section 2.

5. An accuracy assessment chapter is needed. Please, add a description of statistical methods.  The spectral-based results were correlated to the chemical elements' abundance and mineral composition as obtained from chemical analyses. The current study is not based on statistical models (please see lines 45-50 in the introduction).

6. Please, present a variability of acquired spectral properties, e.g. statistical data. ± Stdev. curves were added to figure 1b as an example. What do present the lines (average, median)? A sentence “Each spectrum is the average of tens of pixels in the calculated emissivity image” was added to the legend of figures 2, 3.

7. Please, describe in Methods number of acquired samples (line 53 “Ninety soil samples…”), analyses (lines 55-58), why you have different colors? Different colors represent different soil types, as mentioned in the figures legends.

8. Discussion should contain a comparision of acquired results with achievements of other researchers, so please, compare more important results. You can add a table, to have better comparision. Please, add references to the discussion. Please see our response to comment 2.

Reviewer 3 Report

The work titled " Mineral Classification of Soils Using Hyperspectral Longwave Infrared (LWIR) Ground-Based Data " is an interesting research. This study clearly advances knowledge, and the analysis and presentation of data is also novel. However, there are some major flaws in the manuscript. Overall, from my point of view, the manuscript needs at least major revision before it can be considered for publication. I attach several suggestions and recommendations that could guide the authors through the revision process.

Introduction

Add few more literatures to state the objectives of the work and provide an adequate background.

Material and methods

1.      The Equation in Line 70 should give a reference and basis.

2.      Figure 1 needs to be improved

Results and discussion

1.      All Figures need to be improved.

2.      More scientific reasoning is needed to improve the quality of the work when discussing the results. For example, Line 87-94, why a triplet-like absorption feature with minima at 8.21, 8.85 and 9.33 µm (Figure 2a) indicates the presence of both quartz and clay minerals. Please give a more detailed explanation of the analysis of this part, such as citing references or indicating that the bond corresponding to the wavelength is related to the corresponding substance.

3.      Line 112-113. In general, a larger Al/Si value in the sample indicates a higher concentration of clay minerals than quartz. Ca indicate the concentration of carbonates). More scientific reasoning is needed.

4.      Line 126. Please explain these two indices in detail.

5.      Mark the soil type in Figure 7.

6.      A detailed analysis and comparison should be made according to the results of Figure 7, Table 4, and Figure 8, not just a simple explanation.

Conclusions

Line 164The classification fit the results of the XRD and elemental analyses. Please give specific data for explanation.

Author Response

The authors would like to thank the reviewer. Our response to the reviewer's comments are in red color.

Introduction

Add few more literatures to state the objectives of the work and provide an adequate background. The type of the manuscript is “Letter and as such, includes a brief introduction explaining the concept of the study.

Material and methods

1.  The Equation in Line 70 should give a reference and basis. A citation to reference [19] was added.

2.  Figure 1 needs to be improved. If the comment refers to figure 1a- the image is as acquired with the LWIR sensor.

Results and discussion

1. All Figures need to be improved. The figures look clear to us. We cannot respond correctly without specific comments.

2. More scientific reasoning is needed to improve the quality of the work when discussing the results. For example, Line 87-94, why a triplet-like absorption feature with minima at 8.21, 8.85 and 9.33 µm (Figure 2a) indicates the presence of both quartz and clay minerals. Please give a more detailed explanation of the analysis of this part, such as citing references or indicating that the bond corresponding to the wavelength is related to the corresponding substance. Please see line 95 and the added Figure 4.

3. Line 112-113. In general, a larger Al/Si value in the sample indicates a higher concentration of clay minerals than quartz. Ca indicate the concentration of carbonates). More scientific reasoning is needed. Please see lines 58-60: “The abundance of silicon (Si, dominant in quartz and clay minerals), aluminum (Al, dominant in clay minerals) and calcium (Ca, dominant in carbonates)…”

4.  Line 126. Please explain these two indices in detail. The indices were explained in lines 131-135.

5.  Mark the soil type in Figure 7. Done (Figure 8 in the revised manuscript).

6.  A detailed analysis and comparison should be made according to the results of Figure 7, Table 4, and Figure 8, not just a simple explanation. These figures and table exemplify the correlated text pages 8-10.

Conclusions

Line 164The classification fit the results of the XRD and elemental analyses. Please give specific data for explanation. Lines 170-174 were rewritten.

Round 2

Reviewer 3 Report

No further questions.